# A Portrait of the Rights of Children with Disabilities in Nigeria: A Policy Review

**DOI:** 10.3390/ijerph20216996

**Published:** 2023-10-30

**Authors:** Rose Uzoma Elekanachi, Keiko Shikako, Laurie Snider, Noemi Dahan-Oliel

**Affiliations:** 1Faculty of Medicine and Health Sciences, School of Physical and Occupational Therapy, McGill University, Montreal, QC H3A 0G4, Canada; keiko.thomas@mcgill.ca (K.S.); laurie.snider@mcgill.ca (L.S.); noemi.dahan@mcgill.ca (N.D.-O.); 2Centre for Interdisciplinary Research in Rehabilitation (CRIR)|MAB-Mackay, Montreal, QC H3S 1M9, Canada; 3Shriners Hospitals for Children, Montreal, QC H4A 0A9, Canada

**Keywords:** policy, review, children, disability, human rights, Nigeria, low/middle-income countries

## Abstract

Worldwide, 200 million children experience disability, with the vast majority living in low- and middle-income countries. The United Nations Convention on the Rights of the Child (CRC) places great importance on the rights of all children for the opportunities for survival, growth, health, and development. A subsequent document, the UN Convention on the Rights of Persons with Disabilities (CRPD), identifies children with disabilities as rights bearers who should be considered in all policies and programming worldwide. Nigeria, in 1991 and 2010, ratified the CRC and the CRPD, respectively. Nonetheless, knowledge of the extent to which their disability and child-directed policies considers these two key conventions, in ensuring that children and children with disabilities have access to care within their right remains limited. This study examined the extent to which Nigeria’s current disability and childhood policies have integrated the two child and disability related conventions from the UN. Using a structured search of databases and Nigerian federal and state government websites, we conducted a policy review to identify their disability and child-related disability policies. We also included the CRC and CRPD reports submitted by the Nigerian government to the United Nations Office of the High Commissioner for Human Rights (OHCHR) (2008 and 2010 cyclical year). A thematic analysis, based on the CRC and CRPD report, identified the following six themes: participation, support systems, awareness raising, factors associated with adherence to the CRC, laws and rights, and services. The review showed that the available Nigerian disability policies were federal, with some state policies which aligned with the CRC and CRPD. Also identified was the lack of disability policies specific to children and their families. We concluded that, to ensure proper inclusion of the rights of all children, including those with disabilities, in Nigeria there is a need for a more optimal uptake of recommendations of the CRC and CRPD as laid out by the UN.

## 1. Introduction

Worldwide, an estimated 240 million children live with disabilities (accounting for one third of individuals with disabilities), with the vast majority living in middle- and low-income countries where access to healthcare resources is limited [1,2,3,4]. Notably, access to educational and rehabilitation services is crucial to the success of the inclusion of all children, regardless of their conditions [2], as are exposure to nurturing relationships and positive social norms and beliefs [2]. The United Nations (UN) Convention on the Rights of the Child (CRC) affirms the rights of all children to opportunities for survival, growth, health, and development. A second convention, the UN Convention on the Rights of Persons with Disabilities (CRPD), specifically identifies children with disabilities as rights bearers who should be considered in all policies and programming worldwide.

The countries of sub-Saharan Africa are home to many children with disabilities, many of whom lack access to basic healthcare services [5]. The 2021 UNICEF report shows that most African nations have incomplete records of disability statistics and are further limited to impairment-based prevalence data and largely focus on the most visible forms of functional limitations (e.g., physical disabilities) [6,7]. Nonetheless, the World Report estimates the Disability Prevalence for moderate and moderate-to-severe disability in the population of Africa is 3.1% and 15.3%, respectively [6]. The foremost cause of disability in Africa is infectious and communicable diseases. Considering the abovementioned lack of access to basic healthcare services in many low- and middle-income countries, this creates a worrisome picture as, given adequate access to healthcare services, about 65% of the disabling conditions in childhood are preventable [4]. The Global Population Reference Bureau [8] predicts that, by 2050, the world’s third most populated country will be Nigeria, which currently has a population of over 200 million and has experienced a growing economy over the past decade [8]. Favored by these more optimal economic conditions, Nigeria models its education and health systems for other African countries to follow suit. Hence, Nigeria, known as the “Giant of Africa”, plays a leading role in ensuring compliance with the CRC and CRPD recommendations for childhood disability rights and models proper access to healthcare for other sub-Saharan countries.

Nigeria, the largest country in West Africa, operates under a three-tier governmental structure comprising federal, state, and local levels. Among these three tiers, the federal government holds the foremost authority, providing policy, oversight, and support for the other levels of governance. The state government carries out policies as provided by the federal government, while the local government collaborates with the state to ensure effective policy implementation and equitable resource allocation. Nigeria signed and published its first Disability Act in 1993. More recently, it was a signatory to two UN Conventions that included specific clauses for children with disabilities: (1) the CRC, the most widely ratified international human rights treaty in history, which set out the civil, political, economic, social, health, and cultural rights of children [9], and (2) the CRPD, which is intended to protect the rights and dignity of persons with disabilities [9]. Nigeria signed the CRC on 26 January 1990 and ratified it on 19 April 1991. The CRPD was signed and ratified later, on 30 March 2007 and 24 September 2010, respectively. Both conventions state that children with disabilities have the same rights to healthcare, nutrition, education, social inclusion, and protection from violence, abuse, and neglect as other children. International law obliges countries that ratify these international treaties to actively implement and monitor their implementation into domestic policies and programs. They are also required to comply with the UN in their cyclical review process and submit reports on progressive implementation of the conventions. The process of implementation into domestic policy requires a careful and ongoing assessment of existing policies, institutions, and structures. There have been eight CRC reporting cycles for Nigeria and four reporting cycles for the CRPD since their ratification, as required by the United Nations Office of the High Commission for Human Rights (OHCHR). Of the eight reporting cycles for the CRC, Nigeria has only responded to four cycles (i.e., Cycle 1: 1996; Cycle 2: 2005; Cycle 3 and 4: 2010). These have all gone through the process of review and revision by the OHCHR. Of the four CRPD reporting cycles, Nigeria has only responded to one cycle (Cycle 1: 2012 but reported in 2021). However, this reporting cycle has not been reviewed or received feedback from the OHCHR as only a state report was submitted. According to the OHCHR website, expectations of comparable submissions by other countries to the CRPD include a state party report, list of issues (LOI), and information from civil society.

It is currently known that children with disabilities continue to face violations of their rights when they are denied adequate healthcare, nutrition, education, and protection from violence [9] (OHCHR, 2013). Simeonsson (2000) concluded that ensuring access to appropriate support, such as early childhood intervention and education, can enact the rights of children with disabilities, promote rich and fulfilling childhoods, and prepare them for full and meaningful participation in adulthood [10]. However, the discussion on disability rights frequently focuses on adults, and most child-related initiatives do not consider the specific needs of children with disabilities. Most African countries have policies and legislation related to disabilities but have yet to realize an accessible and equitable continuum of healthcare in rehabilitation [11]. In sub-Saharan African countries, accessible healthcare, and rehabilitation services, when available, are limited to urban areas [5]. Furthermore, the plan of action for the adoption of the CRC and the CRPD was to establish a guiding framework around which national government, international organizations, and non-governmental organizations (NGOs) recognize violations in human rights and develop additional measures to ensure the realization of these rights [2,12]. The intention of the reporting cycles for the conventions is to actively monitor implementation of the human rights treaty, which embraces most aspects of children’s lives. In addition, it evokes a language of rights and entitlement by which to frame indicators as well as national and international mechanisms for scrutiny [13]. 

Several international agencies have proposed strategic plans and normative frameworks to support the elaboration of policies that consider structural aspects supporting health and well-being for marginalized groups in an equitable manner. In particular, the improvement in the developmental outcomes for the participation and protection of children and youth with disabilities has been the objective of a long-term strategic collaboration between the UN, the World Health Organization (WHO), and UNICEF [6,14]. One of the key outcomes of this strategic initiative was the identification of the urgent need to support early childhood intervention for children with disabilities and their families [15]. As itemized in Table 1, the WHO Report on Disability [6] conveyed nine recommendations, such as enabling access to all mainstream systems and services, among others, for the implementation of policy and practice in the field of disability in Nigeria. Policy and service models must be carefully crafted to consider the needs of populations that face multiple levels of marginalization. The development of collaborative approaches across levels of government and sectors is required to positively impact existing barriers to accessing essential services for these groups [11]. A systematic use of normative frameworks to guide policy elaboration and service delivery systems is a promising way to support the reduction in inequities experienced by children with disabilities in Nigeria [11].

In 1993 and 2003, Nigeria established and updated the Disability Decree, which has not been updated since the ratification of the CRC in 1991 and the CRPD in 2010. Furthermore, although recommendations were made by the UN CRPD Committee in 2007, no implementation or monitoring plan has been specified, nor has a specific childhood disability policy been developed. This gap in implementation can be addressed by mapping the existing policies to the CRPD and CRC articles to interpret the current state of implementation of normative frameworks for human rights in childhood disability. A careful examination of existing reports to the UN Conventions, including federal and national policies regarding the rights of children with disabilities, may inform advocacy efforts, rights promotion, and policy development for children with disabilities and their families. Therefore, the purpose of this policy review was to examine the extent to which Nigeria has integrated the United Nations’ CRC and CRPD objectives into its current childhood disability policies. Specifically, this policy review (1) identifies the Nigerian national and subnational policies on childhood disability and (2) examines these policies within the context of the country’s obligations to the UN CRC and CRPD.

## 2. Materials and Methods

### 2.1. Knowledge Synthesis Design

We conducted a policy review to identify Nigerian policy documents, government reports to the UN CRC and CRPD committees (i.e., state reports, LOI, civil society reports, concluding observations (COs)) and the Nigerian federal and state policies on disability, disability rights of the child, and rights of the child. We used the PRISMA ScR Checklist for Systematic Reviews and Meta-Analysis Extension for Scoping Reviews [16] to document search strategies and ensure quality, comprehensiveness, and methodological rigor, as there is little methodological guidance specific to policy reviews. 

### 2.2. Search Strategy

The research team, the first author, and an information scientist developed and conducted a systematic search in policy-related databases and websites (Table 2) using the following keywords: Nigeria, childhood, disability, policies, and health. We carried out this search to identify Nigerian federal and state policies as well as reports to the CRC and CRPD for the most recent review cycle submitted to the OHCHR website in English. For gray literature (i.e., government websites), we searched the first 10 webpages with a date range of 1993–2021. We contacted relevant agencies and libraries by email to request policy documents falling within the scope of our research that were unavailable online. Many Nigerian policy documents were also inaccessible and unavailable online. Only documents which were accessible and met the inclusion criteria (Table 3) were included in this review.

### 2.3. Document Selection

Policies identified in journals, symposiums, conferences, and summits were included in this study if they were available online. Unpublished reports and policies (i.e., internal documents or seed documents), if they were not identified on the website and could only be accessed by contacting the agencies directly, were not included for rigor on the publicly available document criteria. The disability policy documents and reports (i.e., state reports, LOI, civil society reports, summary reports, additional summary, reply to LOI, and COs) included were national and subnational policies related to disabilities and children. These included document and reports identified in policy databases, official government websites, press releases, and Nigerian government reports to the UN CRPD and CRC committees publicly available on the OHCHR website. Policies and reports identified though the search were reviewed according to the selection criteria by one reviewer. The list of included and excluded reports was discussed with the other authors in regular meetings for agreement.

### 2.4. Coding Framework 

We developed an initial coding framework based on the UN CRC and CRPD articles. We also used a text-mining tool (WordStat V.9) to apply an initial analytical categorization model developed based on the CRPD and CRC articles addressing the rights of children with disabilities [17]. The Wordstat categorization model was used to text-mine words and themes related to the CRC and CRPD according to frequency. CRC and CRPD articles that had 80% frequency and above of text identified through the categorization model (i.e., Articles 20, 23, 28, and 31 of the CRC and Articles 7, 8, and 23 of the CRPD) were included to form the initial coding framework in NVivo. All the documents were then uploaded into NVivo for thematic framework analysis [18]. Hierarchical coding was used to establish broad themes (parent nodes) that encompassed successively narrower, more specific subthemes (child nodes) that exemplified different aspects, types, and interpretations of the major themes (parent nodes). 

### 2.5. Analysis and Synthesis

The initial coding framework generated through the keywords and themes on Wordstat was created in NVivo and pilot-tested by the first author (RE) on one report and one policy document. The research team (RE, KS, NDO, LS) met to discuss the emerging themes, identify nuances, and discuss areas that should be re-classified into existing themes (i.e., CRC or CRPD articles identified in the WordStat analysis) or if new themes should be created. The first author (RE) proceeded with coding the included documents with regular discussions with the research team (RE, KS, NDO, LS) on any new or diverging theme identified. Every third Nigerian policy document was coded by two members of the research team (RE, NDO) during bi-weekly meetings to review and validate the coding process, ensure agreement, and address discrepancies on new emerging themes. Analysis of the codes was performed by two members of the research team (RE, NDO) and critically debriefed with the other members of the research team (KS, LS). A coding book was kept, ensuring proper tracking of the coding process on NVivo. Every analytical step and decision were documented to inform the data analysis and results of the policy review. A descriptive analysis and a qualitative deductive thematic synthesis using the coding framework were used to analyze the findings.

### 2.6. Researchers’ Position Statement

Walt et al., 2008 mention that “researchers rarely reflect on how their own positions shape their interpretations and conclusions in a policy review” [19] [p. 309]. Therefore, as first author (RE), I will outline the Nigerian national and subnational policies on childhood disability and examine these policies within the context of the country’s obligations to the UN CRC and CRPD. It is important that I state my position in order to ensure transparency. I am a Nigerian physical therapist with three years of clinical experience working with children with disabilities. I am also a PhD candidate with an interest in childhood disability and disability policy research. Hence, experiencing the challenges caregivers face in accessing care for their children with disabilities in my home country Nigeria, along with the financial constraints involved in caring for their child, shapes my perspective. I will be reflecting on my experiences as a physical therapist, researcher, and student in the field of policy review and childhood disability research as I discuss the current disability and childhood disability polices in Nigeria and how they reflect the UN CRC and CRPD conventions and their recommendations with the expertise and methodological guidance from the other members of the research team (KS, NDO, LS).

## 3. Results

### 3.1. Search Strategy and Policy Document Selection

The search identified a total of 26 documents, of which 13 met the criteria for inclusion (Table 3). Included were eight Nigerian CRC government reports from the most recent reporting cycle (i.e., a combined report for the 2008 and 2010 cyclical years) of the CRC and six other Nigerian disability and childhood disability policy documents. No complete CRPD reports were found at the time of the search. Therefore, in the discussion of the results, all the Nigerian CRC reports will be referred to by their titles as no CRPD reports were included in this review. The same will apply to the policy documents. Table 3 outlines the inclusion and exclusion criteria. Table 4 lists the identified documents and reasons for exclusion. 

### 3.2. Analysis and Synthesis 

The CRC and CRPD articles that produced the highest keyword frequency (Table 5) in the text mining procedure and were used to develop the initial coding framework in NVivo are depicted in Table 6. In this section, the articles and policy documents are referred to using the original document name and year of publication to ensure clarity. Using these articles as a guiding framework, six analytical themes were identified, as shown in Table 7, with their associating document and reports. They included (1) participation, (2) support systems, (3) awareness raising, (4) factors associated with adherence to the CRC, (5) laws and rights, and (6) services. An outline of the themes and subthemes in this policy review is presented in Appendix A. 

### 3.3. Theme 1: Participation

This addresses the policies, laws, and programs developed to integrate children with disabilities into the community in the areas of education, recreation, and vocational development, and those ensuring that the voices of children with disabilities were included when developing childhood disability policies or laws. Two subthemes were identified: (i) civil participation and (ii) community participation. 

Civil participation focuses on the involvement of children with disabilities and their families in policy development, advocacy, and other civil rights related to disability. It is important to note that because there were no complete Nigerian CRPD reports included in this policy review, not much aligned with the CRPD. Nonetheless, CRPD Articles 7, 8, and 23 mandated the existence of a comprehensive national disability awareness-raising strategy aimed at combating stereotypes against persons with disabilities and promoting awareness of their rights, which is well corroborated in the awareness raising theme. The CRC reports emphasized the rights of children and young people to know, express, and promote their opinions in any process involving them as guaranteed under the Child Rights Act (CRA), 2003 (Additional Summary Record, 2010), and through the establishment of child rights clubs in schools (State Party Report, 2008). These child rights clubs incorporated the relevant provisions of the CRC. The 2010 CO welcomed the establishment of the Children’s Parliaments in all 36 states, in line with the CRC recommendations, to ensure that children were consulted in the process of budget allocation and their active participation in international and national forums such as the Day of the African Child, etc. (State Party Report, 2008). The Parliament was institutionalized to provide a platform for children to dialogue with leaders with a mandate to represent the voices and aspirations of the Nigerian child, while advocating for their survival, protection, development, and participation regarding their own rights (State Party Report, 2008). However, despite this effort at inclusion, it is unclear if children with disabilities are designated members of the Children’s Parliament and have a play in actual decision making. In 2008, the State Party Report noted that participation opportunities for children in matters concerning their rights and welfare had progressively increased over the years since the ratification of the CRC and establishment of the Children’s Parliament. However, the CRC committee still expressed concerns about the limited participation of children in matters affecting them in children’s institutions of all kinds (family, community, judicial, and administrative procedures).Community participation covers aspects related to participation in different areas of community life (e.g., access to the community; physical accessibility of public spaces). For instance, the Lagos State Disability Bill (2010) issued building codes and directives for accessible design, such as requirements for lifts and ramps [21]. A five-year transition period from the date of the law’s commencement was given for all public buildings, roads, pedestrian crossings, and all other structures to be modified, accessible, and usable by persons living with disability. The Disability Decree (1993) assured structural adaptation of all educational institutions to the needs of persons with disabilities, guaranteeing physical accessibility of public institutions and facilities [20]. However, although children with disabilities would benefit from the measures outlined in the Disability Decree in public spaces and services, there were no specific mentions of the needs of children identified other than the provision of these structural adaptations. The decree also asserts that “It is the responsibility of all organs in the Federal Republic of Nigeria to provide for the disabled; access and adequate mobility with its facilities and suitable exits for the disabled” [20] [p. 4]. The Reply to LOI (2010) notes that, in some states, children with disabilities have access to scholarships, free medical care, school buses, recreation facilities, book subsidies, and other accessibility equipment. Both the Nigerian Child’s Rights Act (CRA) (2003) and the Lagos State Disability Bill (2010) aim to establish the rights of children to access training, healthcare, rehabilitation services, and general improvements for persons with disabilities [21,23].

### 3.4. Theme 2: Support Systems

This theme describes the framework needed to provide support for children and families of children with disabilities. Two subthemes were identified: (i) emergency and medical support and (ii) structural and financial support.

Emergency and medical support addressed the assistance or essential support for basic needs provided in the context of childhood disability to children and their families as a temporary aid in times of crisis or difficult circumstances. The Disability Decree (1993, p. 5) asserted “the responsibility of the government to provide for the disabled such as acquisition of prosthetic devices and specialty services; a program to assist the families of the disabled to adjust to disability etc.”. The decree also noted that people with disabilities are to be provided free medical and health services in all public health institutions. The 2008 State Party Reports identified notable action points on support provided to persons with disabilities. These notable actions included the establishment of the Emergency Preparedness and Response Project (EPR) for children in situations of emergency (State Party Report 2008, p. 120). Other examples include, but are not limited to, the Adamawa State Maternity Assistance to Women and Child Care Law (2001), Bauchi State Hawking by Children (Prohibition) Law (1985), etc. (State Party Report, 2008). The immunization and food fortification program were part of an annual government plan to effectively detect, control, and eliminate outbreaks of diseases affecting child health, growth, and development, with disease causing more impairments and disabilities in children receiving more attention.Structural and financial support referred to help provided by the government or other establishments to children with disabilities and their families. The Lagos Disability Bill (2010) established a Lagos State Persons Living with Disability Fund to advance the cause of persons living with disability in the state. Steps to ensure the self-reliance of persons living with disabilities and the provision of adequate assistance to persons living with disabilities who desire to be employed were noted to have been followed through by the government. In 2010, the Lagos State Disability Bill clearly directed the government to take steps to ensure the self-reliance of persons living with disability and accordingly give adequate assistance to those who desire to be employed [21]. The CRA (2003) asserted that the services provided by a state government must include conditional or unconditional assistance given in kind or, in exceptional cases, in cash [23]. Despite the support system efforts by the Nigerian government, the committee report in the CO (2010) recommended that the state party take all necessary measures to ensure the allocation of appropriate support programs that assist parents, especially single mothers and teenage households, or legal guardians in the exercise of their responsibilities (CO, 2010). However, there was no mention of provision or support specific to children with disabilities.

### 3.5. Theme 3: Awareness Raising

This theme incorporated education and sharing knowledge about the rights of children with disabilities. Two subthemes were identified: (i) public-at-large awareness raising, which addresses the general awareness of individuals and the community about disabilities and the rights of children with a disability and (ii) government-led awareness activities, which consist of activities, policies, and initiatives on raising awareness on topics of childhood disability at the national level. However, both themes were merged following similarities as the government spearheaded the initiatives and created directives for community members to carry out public awareness-raising activities. The adoption of the CRPD has since led to a replacement of terms defining disabilities to “persons with a speech disability”, “persons with impaired vision”, or “persons with a hearing disability” (Additional Summary Record 2010, p. 8). The Summary Report (2010) also confirmed that “the age of majority had not yet been changed to 18 years old in some states and that the federal government was making the best of that situation because it favored dialogue and awareness-raising” (Summary Record 2010, p. 5). The government’s intention was to avoid a clash with the states concerned, so as not to dissuade them from adopting the CRA (2003), which was a short-term priority. However, there was no mention in this context of the age of majority for children with disabilities. The State Party Report also noted that necessary awareness campaigns not explicitly described in the Summary Record (2010) on the age of majority would continue to ensure the transfer of child-related issues into concurrent legislation. Cultural resistance to the CRA (2003) was identified by the State Party Report as an issue. Assistance of the federal authorities was requested in supporting awareness campaigns that involved religious and traditional leaders in the states that had not yet ratified the CRA. The Lagos State Disability Bill (2010) noted that there was a need for the re-orientation and education of the public on an informed attitude towards persons living with disability [21]. 

### 3.6. Theme 4: Factors Associated with Adherence to the CRC 

This theme only encompassed factors associated with adherence to the CRC, as Nigeria had not reported back to the CRPD on its first cycle, which was overdue (October 2012). Recently, Nigeria submitted a state party report to the CRPD (March 2021). However, no LOI report or civil society report was submitted, which means the report has not undergone a review process and hence cannot be included in this review. In 1991, Nigeria ratified the CRC, and, accordingly, no child can be denied the right to healthcare services. Incorporated in this right are three provisions: (1) medical assistance and healthcare for all children, (2) adequate pre- and postnatal care for mothers, and (3) access by all segments of society to basic knowledge of child health, nutrition, environmental sanitation, and child-health-related issues. Since its ratification in 1991, Nigeria has worked with the CRC to ensure these agreements are achieved. Hence, this theme was established to reflect influential or associated factors regarding adherence to the CRC. Three subthemes were identified: (i) Obstacles; (ii) Enablers; and (iii) Needs Met.

Obstacles includes barriers that the Nigerian government faced in implementing the conventions (Appendix A). According to the Additional Summary Record (2010), a proposed study on providing free healthcare for mothers and children was not deemed feasible. However, a new budget line was established later that year for the implementation of a healthcare program for mothers, newborns, and infants. Also identified in the Additional Summary Record was the lack of specific budget resources to children; however, ministries such as the Ministry of Health and the Ministry of Education provided a budget line for child issues. Appendix A describes other issues arising. The CO (2010) outlined the efforts of the state party to raise awareness of children’s rights through training and sensitization programs for critical target groups. On review, the CRC committee recommended training for all professional groups working with and for children, including necessary revisions of training manuals and operative procedures (CO, 2010). However, there were concerns that discrimination against children prevailed, particularly among young females, children with disabilities, unhoused children, and children of ethnic minority groups (e.g., the Ogboni community, in the Niger Delta region of Nigeria). Furthermore, there were concerns about the lack of information following the convention’s recommendations relating to children with disabilities, as no comprehensive policy had been developed. Also noted by the committee in the CO (2010) was the use of offensive and derogatory definitions and categories employed by the state party when referring to children with disabilities. In the Summary Record (2010), regrets were expressed that 12 states had still not adopted the CRA despite the state’s efforts to incorporate the convention into domestic law. The State Party Reports (2008) acknowledged the challenges (religious and civil strife, economic constraints, unemployment, and heavy debt) faced by the country which may have impeded progress on the full actualization of the child rights outlined in the convention as well as other obstacles outlined in Appendix A.Enablers (Appendix A) summarizes the actions taken by the Nigerian government to facilitate implementation of the convention at a systems level. These included efforts made at the federal, state, and local government levels in Nigeria to actualize the provisions of the convention and ensure its effective practical implementation. Following the inauguration of the Children’s Parliament in 2000, all 36 states and the federal capital territory have been reported to have functional parliaments, a measure to ensure respect for the views of the Nigerian child (State Party Report, 2008). Although the committee welcome the establishment of the children’s parliament, there is concern about the limited participation of children in matters affecting them in institutions of all kinds, and the state party has been urged to strengthen the functioning of the Children Parliaments, ensuring proper representation of all children (e.g., orphans, children with disabilities, refugee children, etc.) (CO, 2010). The Additional Summary Record (2010) requested information on the implementation of the principle of non-discrimination. Two examples of actions taken by the government to combat discrimination were cited. The first example was the introduction of “child-friendly schools” in the Nigerian Northern States to encourage female school enrolment. In this context, a “child-friendly school” demonstrated parity between the sexes (Additional Summary Record, 2010, pp. 7–8). However, there was no specification on the availability of this initiative for female students with disabilities. The second example demonstrated the efforts made to raise the profile of prominent women and make them better known in the country’s various regions to promote a positive image of women, offering role models with whom children and the public could identify (Additional Summary Record, 2010). Although these factors relate to vulnerable groups of children, there was no particular mention of children with disabilities. Other convention implementations steps include the timely data collection on issues relating to children with disabilities for more effective intervention, as highlighted in Appendix A, and increased provisions for equal educational opportunities for all children, irrespective of their disability.Needs Met (Appendix A) summarizes the factors that were described as facilitating the implementation of the conventions, where the needs of children with disabilities were met by the Nigerian government. The Summary Record (2010) also noted the formulation of the National Child Policy with clear objectives to establish an environment conducive to the exercising of rights described in the convention. The Nigerian government then developed guidelines on the management and monitoring of childcare institutions, including orphanages, and established family courts in eight states and the Federal Capital Territory to support implementation of the CRA (2003). However, these guidelines did not specify the provision for or inclusion of children with disabilities in the management and monitoring of childcare institutions. According to the State Report, 2008, provisions of the CRA guaranteeing special protections measures for children, and their implementation, have improved with the adoption of the act by many states of the federation.

### 3.7. Theme 5: Laws and Rights 

This theme includes laws in the Nigerian policy documents that reflected mechanisms supporting the legalization of rights, such as penalties and complaint mechanisms. The subthemes identified were (i) human rights laws and (ii) fundamental rights. 

Human Rights Laws (Appendix A) include all the Nigerian policies and laws relating to childhood disability. The 1993 Disability Decree clearly noted that “disabled persons shall have equal rights, privileges, obligations, and opportunities before the law” [20] [p. 1]. In relation to children, the CRA (2003) states that every child has a right to survival and development and, where a caregiver with custody of a child fails in their duty, they are liable on first conviction to a reprimand conviction with community service and on second conviction to a fine of NGN 2000 or imprisonment [23]. Additional information on the Reply to LOI (2010) highlighted the plan of the government to ensure the protection of the most vulnerable children through improved policy and legislation. The Disability Decree (1993) also stated that all persons with disabilities will be guaranteed equal treatment, and the government will be responsible for adopting and promoting policies to ensure the full integration of persons with disabilities into the society [20].Fundamental rights include the right to clean food, water, clothing, and an overall good standard of living (welfare, social development, poverty reduction) for children with disabilities and their families. The CO (2010) noted the amendment to the constitution with a view to guaranteeing the right of the child to the best attainable state of physical and mental health as a constitutionally protected right. This was implemented with a view to specify the respective powers and responsibilities of federal, state, and local governments in the delivery of healthcare. The Lagos State Disability Bill (2010) stated its responsibility for the receipt of complaints from persons living with disability on the violation of any of his or her rights, actualizing the enjoyment of all rights in this law by persons living with disability [21], as detailed in Appendix A. Accordingly, the government shall take all appropriate measures to ensure that the rights of persons living with disabilities are guaranteed [21].

### 3.8. Theme 6: Services

The CRA (2003) stated that every person, authority, and institution responsible for ensuring the care of a child in need of special protection measures shall endeavor to provide the child with such assistance (education, employment, rehabilitation, and recreational opportunities) to allow them to achieve the fullest possible social integration [23]. According to the Reply to LOI (2010), federal and state governments, alongside NGOs and religious organizations, have been active in the provision of services for the children with disabilities with measures and policies to protect their rights. Hence, this theme addresses the policies on services provided for children with disabilities and their families. Eight subthemes were identified: (i) Childcare and Family Support, (ii) Transportation, (iii) Social, (iv) Vocational, (v) Special Care and Programs, (vi) Education, (vii) Healthcare, and (viii) Institutional and Alternative Care.

Childcare and Family Support: The Disability Decree (1993) stated that the government would ensure that policy guidelines for housing took into consideration the needs of the disabled [20]. The government was mandated to subsidize accommodation for persons with disabilities, apportion not less than 10% of all public housing to this group, improve the accessibility of existing housing facilities, and provide assistance with childcare services. Postal agencies were mandated to provide persons with disabilities with free postal services for all materials to support access to learning aids and orthopedic devices. The CO Report (2010) recommended that the state party take all necessary measures to ensure the allocation of appropriate financial and other support to allow caregivers to exercise their responsibilities. The main goal of the NPA was to ensure that improved quality programs and services are implemented for the protection, care, and support for vulnerable children and their caregivers through improved policy and legislation (Reply to LOI 2010). Sections 16 (2) (d) and 17 (3) of the Nigerian constitution recognize children with a physical and emotional disability as a vulnerable group that needs to be supported financially, materially, and technically and protected from all forms of exploitation and abuse. At the state level, the government is mandated to undertake the provision of early and comprehensive information, services, and support to children with disabilities and their families with no separation of a child from their parent based on a disability of either one or both parents [21].Transportation includes services offered by the government through policies for children with disabilities and their families. The Disability Decree (1993) noted that a disabled person was entitled to free transportation by bus, rail, or any other conveyance (other than air travel) that serves the public [20]. Public transport systems are required to adapt vehicles and make them accessible for needs of the disabled with priority granted in all publicly supported transport system and a reasonable number of seats reserved solely for the use of persons with disability. Subsequently, the Lagos State Disability Bill (2010) noted that, to ensure the convenience and safety of a person with a disability, all transport service providers should make available, and mark appropriately, 1 out of every 10 seats in vehicles, vessels, trains, or aircraft for the use of persons with a disability [21].Social is associated with the social services provided for children with disabilities and their families. The Disability Decree (1993) on supportive social services stated that the government will provide appropriate auxiliary social services to persons with disabilities [20]. It was also noted that assistance would be rendered in all ways appropriate (e.g., prosthetic devices and medical specialty services, specialized training activities, follow-up rehabilitation services, counselling, self-image programs) for the families of children with a disability. In CO (2010) the UN suggested that Nigeria should ensure strategic budgetary lines for disadvantaged or vulnerable children (e.g., orphans, homeless children, internally displaced children) requiring affirmative social measures (i.e., birth registration and protection), even in situations of economic crisis, natural disasters, or other emergencies.Vocational addresses skills training for different types of jobs, volunteer (unpaid jobs) and employment opportunities (actual placement and adaptations) and programs for children with disabilities. The Disability Decree (1993) advocated vocational training to develop skills [20]. Accordingly, the government received a mandate to promote the employment of persons with disabilities by establishing centers, training programs, vocational guidance, and counselling to develop and enhance the skills and potentials of persons with disability. However, it was not clearly stated how this applies to children or youths with disabilities. The Reply to the LOI (2010) noted that various governmental and non-governmental organizations operated vocational training centers, special schools, and homes for children with disabilities in different parts of the country. The Lagos State Disability Bill (2010) noted the establishment and promotion of schools and vocational and rehabilitation centers for the development of persons living with disability, with training institutions to facilitate the acquisition of special skills by persons living with disability [21].Special Care and Programs refer to care, or programs organized for children with disabilities by the community or agreed upon by the government. The State Party Report notes that “except for the provision made in the rehabilitation centers, there are no general specialized services for physically challenged children” (State Party Report 2008, p. 89). However, the cumulative effect of the CRA, 2003 (Sections 11, 13 and 16), guarantees the rights of physically and emotionally challenged children to dignity, self-reliance, and active participation in the affairs of the community (State Party Report, 2008). The Reply to LOI (2010) identified measures to protect the rights of children with disabilities. Most states in Nigeria provide children with disabilities with support gadgets (e.g., crutches, wheelchairs, tricycles, hearing aids, braille machines) to facilitate their development, as well as special sports designed for their convenience and active participation (Reply to LOI Additional Information 2010).Education comprises the educational services offered by the Nigerian government for children with disabilities. According to the 1993 Disability Decree, a National Institute of Special Education was established to cope with the increasing research and development in the field of education for persons with a disability [20]. The decree outlined the government’s plan to establish special schools with appropriate curricula for disability conditions, ensuring in-service training of teachers [20]. It also mandated the government to ensure that not less than 10% of all educational expenditures are committed to the education needs of persons with disabilities at all levels. According to the 2003 CRA, every parent or guardian is to ensure that their child or ward attends and completes primary, junior, and secondary education [23]. The decree mandated that persons with a disability are provided equal, free, and adequate education in all public educational institutions at all levels, and that all schools (pre-primary, primary, secondary, and tertiary) be made accessible, with trained personnel available to address their educational development. Furthermore, the needs and requirements of persons with disabilities should be considered in the formulation, design of educational policies and programs, and promotion of specialized institutions to facilitate research and development [20]. The Lagos State Disability Bill (2010) mandated the government to establish special model schools for persons with disabilities to ensure education was delivered in the most appropriate languages, modes, and means of communication to maximize academic and social development [21]. The bill asserted that children with disabilities should have equal access to participation in play, recreation, leisure, and sporting activities in the school system [21].Healthcare is associated with the healthcare services provided to children with disabilities. As stipulated in the CRA (2003), the National Health Bill was the government’s answer to financing healthcare in order to provide a direct funding line for primary healthcare, guarantee the rights of the child to the best physical and mental health, ensure the provision of free maternal and child health services, and take measures to ensure nation-wide implementation of the 2005 National Health Insurance Scheme [21]. For example, the Lagos State Disability Bill (2010) noted that “any hospital where a person living with a communication disability is being attended shall ensure provision for special communication equipment” [21]. The committee welcomed the information that stated that investment in children’s programs and budget allocations to health and education had increased (CO, 2010). The committee strongly recommended the State Party to “continue its effort to ensure access to education and health services in all states for all children with disabilities and address existing geographical disparities with respect to available social services” (CO 2010, p. 14). The Lagos State Disability Bill (2010) also guaranteed persons living with disability unfettered access to adequate healthcare (free medical and health services in all public health institutions) without discrimination on the basis of disability [21].Institutional and alternative care refers to services or care offered by the government for children with disabilities and their families in different forms of institutions. In the CRA (2002), the federal government encouraged private organizations to provide services for children in need (e.g., children with disabilities) as permitted by the government [23]. The act also reiterated the government provision for the reception and accommodation of children who had been removed from family or kept away from home under Part IV of the Act (protection of the children); however, it was not clearly stated to what extent this applied to children with disabilities. The CO (2010) stated that “effective measures to ensure that the rights of a child are heard, respected, and implemented in all civil and penal judicial proceedings as well as in administrative proceedings, including those concerning children in alternative care” (p. 8). The Lagos State Disability Bill (2010) noted that the office would provide and sponsor alternative care for a child living with disability [21].

## 4. Discussion

The aim of this policy review was to comprehensively (1) identify the Nigerian national and subnational policies on childhood disability and (2) examine these policies particularly within the framework of the UN CRC and CRPD. The analysis involved an in-depth examination of eight distinct reports (state party reports, statements, summary records, additional summary records, LOI, reply to LOI, COs) submitted to the UN CRC Committee and six key Nigerian childhood disability policy documents. The review identified that while the majority of these documents addressed issues concerning children or individuals with disabilities, they offered limited explicit provisions and indicators specifically tailored to children with disabilities.

The International Classification of Functioning, Disability, and Health (ICF) [26] defines disability as neither purely biological nor social but, rather, as the interaction between health conditions, environmental factors, and personal factors. Similarly, the social determinants of the health framework and the ICF advocate for a comprehensive perspective on health as a fundamental right [27]. The ICF framework asserts that disability occurs at three levels: as an impairment in body function or structure, a limitation in activity, or a restriction in participation. In parallel to the mapping process of the Nigeria policy documents and reports to the CRC and CRPD articles, these conventions were considered as complementary rights documents, as supported by Underwood et al., 2018, providing a framework to guide and monitor the rights of children and children with disabilities while shaping healthcare services, programs, and initiatives in Nigeria [28]. However, the policy documents and reports from Nigeria displayed limited considerations specific to children with disabilities (i.e., play, community participation, evolving civic capacity), revealing a notable aspect of intersectionality that leaves room for neglect in policy development.

This policy review identified several essential components forming a scaffolding structure for the rights of children and children with disabilities in Nigeria. These components encompassed the adoption of legislation enacting the Child Rights Act of 2003 in 24 states of the federation, including policies and strategies to bolster the convention’s implementation (e.g., National Child Policy, National Child Health Policy). Also included was the National Policy Action, which emphasizes the prioritization of children, health education, the protection of children (CO, 2010, p. 3), and the appointment of a special rapporteur on child rights, which is mandated to monitor and collect data on violations of children’s rights.

Crucial actions identified (Appendix A) through this policy review included increased budget allocation to health and education, investments in children’s programs, training initiatives, and child rights awareness sensitization programs, etc. However, most of these initiatives are policy innovations. Other policies identified also highlighted ongoing surveys to gather information on children with disabilities and the provision of special education facilities for them. Yet, a comprehensive policy specifically targeting children with disabilities was notably absent, as noted by the UN committee. The lack of attention paid to children with disabilities during the implementation and enactment of these conventions [27] further deepens the gap between services and systems offered to individuals and children with disabilities. A recent review of disability issues in Nigeria identified various factors hindering notable progress, including the absence of disability discrimination laws, inadequate social protection, limited public understanding of disability issues, and poor access to rehabilitation services [29]. Lang and Upal (2008) recommended in a report, among other considerations, the need for the collection of robust, reliable data and advocacy of the passage of disability bills into law [30].

The mapping process in this policy review identified participation in line with the CRPD (Articles 7, 23). Discrimination regulations outlined by the CRPD were also discussed, particularly concerning the implementation and monitoring of laws, policies, regulations, and programs related to persons with disabilities within family life. While aspects of inclusion of children in civic life were identified in the disability policy documents and reports, a notable gap existed regarding the inclusion of children with disabilities in the Children’s Parliament. The imperative of inclusion and access to education was emphasized in the Nigerian policy documents, encompassing all children, vulnerable children, and children with special needs. However, the processes proposed to ensure true inclusion through implementation and enactment mechanisms were not clearly delineated. Children residing in lower socioeconomic settings face a higher risk of various health outcomes, while children with disabilities also experience lower participation and suboptimal developmental and health outcomes [31,32,33]. Aligning with CRC Articles 20, 23, 28, and 31, the theme support systems outlined that children with disabilities have access to education, vocational training, healthcare services, rehabilitation services, and recreation opportunities to facilitate their social integration and individual development. Services mapped the rights of children with disabilities to different types of services, such as free education, healthcare, and financial assistance for special care, which correspond with CRC Articles 20, 23, 28, and 31. In Nigeria, there are established teaching, orthopedic, and other specialist hospitals, but persons with disabilities have limited access to services by these institutions as provided by the National Health Insurance [34]. As regards to education, there has been advancement in the commitment level demonstrated for children with disabilities in some Nigerian states; however, there are problems with accommodating and providing quality education, such as the limited number of special educators and education units in Nigeria and logistic and financial problems on the part of the government, causing a high cost of special education in Nigeria [34,35].

Furthermore, this policy review brought attention to the discrepancies between the CRC and the CRPD concerning alternative living arrangements for children with severe disabilities [2]. The CRC allows for placement in alternative institutional care, a practice prohibited by the CRPD. Long-term placement in institutions was deemed harmful, particularly exacerbating intellectual disability or causing severe developmental delays among children who were not intellectually disabled [36]. In accordance with the CRPD Article 19, there is a focus on the main elements of choice in individualized support that promotes inclusion, prevents isolation, and ensures for general services’ accessibility for persons with disabilities [37]. Hence, the need for a national deinstitutionalization strategy was emphasized, advocating for community-based services, the provision of support to families, education of healthcare providers and caregivers, and societal awareness [36].

Despite the intended purpose of fostering equitable policy development and providing specific language and indicators for progressive implementation [12], this review highlighted that the ratification of the conventions and the passage and adoption of supportive legal frameworks, while necessary, were not sufficient conditions to guarantee the recognition of children’s rights in Africa [12]. This inadequacy is evident in the lack of attention paid to children with disabilities in other Nigerian policies relating to children or persons with disabilities. Although the Nigerian government ratified the CRC and CRPD, leading to the enactment of several laws intended to address the rights of persons with disabilities and protect their interests [34], these laws did not specifically address children with disabilities. Also emphasized was the critical role of civil society, including non-governmental bodies, national human rights institutions, and individual experts, in monitoring and implementing the conventions [38].

Itulua-Abumere, 2013, outlines that the main reason Nigeria seems to be behind in the safeguarding of children with disabilities is the lack of acceptance of social work and welfare service by the Nigerian society [38]. The complete acceptance, financing, and incorporation of social and welfare services to help persons with disabilities access their necessary needs and transfer information to relevant bodies are imperative [38]. Therefore, there is a need for the implementation of awareness and training programs by the government (at federal, state, and local levels). This will help to increase the Nigerian population’s acceptance rates of these services, which may lead to the progressive implementation of comprehensive disability policies. 

The engagement of civil society organizations was underscored as crucial for enhancing domestic advocacy and enabling a more inclusive and comprehensive national dialogue to improve the realization of socioeconomic rights. The lack of civil society reports for Nigeria was recognized as a challenge, hindering a comprehensive assessment of the extent to which the state report aligned with the lived experiences of Nigerians. This is the case in South Africa, as a review of its reporting obligations to human rights bodies and mechanisms showed that effective civil society involvement in the reporting process is lacking [39]. As exemplified in Viljoen and Orago’s 2014 study, civil society organizations are known to help enhance domestic advocacy through the widespread dissemination of any material or document emanating from a human rights treaty, with the effect that the national dialogue will be more inclusive and comprehensive, improving the realization of socioeconomic rights [40].

## 5. Limitations and Implications for Future Research 

This policy review is the first to address the policy profile of childhood disability in Nigeria, providing insights into disability policy status. In this policy review, most of the disability policies which were readily accessible through database and website searches were federal-related bills and just one state-related policy (i.e., Lagos State Disability Bill, 2010). Despite later findings of state-related policies in CRC reports (Appendix A), the content of these documents remains inaccessible for review. Although the CRC and CRPD articles were used to form the mapping framework, this review was unable to map the Nigerian CRPD reports due to unavailability. This lack of reports and access to policy documents may hinder policy implementation and alignment with UN goals, particularly for children with disabilities, underscoring the pivotal advocacy role of health and education professionals serving these families [27] At the time of this policy review, there has been no updated cycle report from Nigeria since 2015 and no recent updates to the different policy documents.

This jurisdiction of control and services between the federal and state government could also be a contributing factor to this review’s limitations. Therefore, future research should identify and collaborate with civil society organizations, families of children with disabilities, youth advocacy organizations, public servants, and government officials to validate the implications and applications of available policies; identify barriers to the implementation of the UN conventions and specific mechanisms for policy development and collaborations; bridge the gap between disability and child policies to contemplate the needs of children with disabilities and ensure they are not left behind in policy and program development; and lastly, identify efficient monitoring and reporting mechanisms that are contextualized to Nigeria and can serve as a model for other states.

## 6. Conclusions

In conclusion, this policy review emphasizes the pressing need for a targeted, comprehensive policy approach addressing the rights and needs of children with disabilities in Nigeria. Recommendations include heightened awareness programs, enhanced policy accessibility and dissemination, and active civil society involvement in monitoring and reporting to the CRC and CRPD. Addressing these gaps is vital for fostering a more inclusive and equitable society for Nigerian children with disabilities.

This review confirms the presence of disability policies in Nigeria but reveals gaps in their implementation. These policies, which primarily operate at the federal level with some state alignment to the CRC and CRPD, lack specific focus on children with disabilities and their families. Hence, bridging this gap necessitates policies aligning child disability rights with societal needs, ensuring effective policy implementation. Articles 7, 8, and 23 of the CRPD articles offer recommendations for comprehensive childhood disability policies. Addressing policy gaps, encouraging timely implementation, and aligning with international recommendations are imperative for the proper inclusion of children with disabilities and their families.

## Figures and Tables

**Table 1 ijerph-20-06996-t001:** World Health Organization recommendations.

	Recommendations
1.	Enable access to all mainstream systems and services
2.	Invest in specific programs and services for people with disabilities
3.	Adopt a national disability strategy and plan of action
4.	Involve people with disabilities
5.	Improve human resource capacity
6.	Provide adequate funding and improve affordability
7.	Increase public awareness and understanding
8.	Improve disability data collection
9.	Strengthen and support research on disability.

**Table 2 ijerph-20-06996-t002:** Databases and websites search.

Databases and Websites	Professional Body Websites
Goggle Scholar (accessed on 20 April 2020)	Office of the High Commissioner United Nations Human Rights
www.hrw.org (accessed on 20 April 2020)	World Health Organization
www.ohchr.org (accessed on 20 April 2020)	Global Health Workforce Alliance
www.unicef.org (accessed on 20 April 2020)	Global Health Data Exchange
www.policyproject.com (accessed on 20 April 2020)	United Nations Children’s Fund
https://dredf.org (accessed on 20 April 2020)	Federal Ministry of Health Nigeria.

**Table 3 ijerph-20-06996-t003:** Inclusion and exclusion criteria for selection.

Inclusion Criteria	Exclusion Criteria
Nigerian Policy Documents	
Policy documents addressing disability in Nigeria	Documents that did not address disability policies in Nigeria
Policy documents addressing children and child-related health and disability in Nigeria	Drafts and unapproved policy documents
Policy documents approved and publicly accessible by an online search and website of the Nigerian respective organization	
Nigerian CRC Reports	
Complete reports to the UN CRC	Incomplete reports to the UN CRC and CRPD
Complete reports to the UN CRPD	Reports not reviewed or replied to by the UN CRC and/or CRPD committees
Recently submitted reports to the UN CRC and CRPD	

**Table 4 ijerph-20-06996-t004:** Search results of policy documents included and excluded from this review.

Policy Document	Decision	Reason for Decision
Nigerian Disability Decree Act (1993) [20]	Included	First approved and published disability act in Nigeria. Addresses disability in Nigeria with a section addressing children.
Lagos State Disability Bill (2010) [21]	Included	This is a bill that concerns individuals with a disability. A state policy in Nigeria.
National Child Health Policy (2006) [22]	Included	This policy document addressed the protecting and promoting of the health of children.
Nigerian Child Rights Act (2003) [23]	Included	This policy document concerns children. Federal policy in Nigeria.
Discrimination against Persons with Disability Act (2010) [24]	Included	This disability act addresses discrimination against individuals with a disability.
Convention of the Rights of the Child Report (2010) [25]	Included	This report included the complete 8 report documents (state party report, LOI, delegation list, summary record, additional summary record, reply to LOI additional information, CRC statement, and concluding observations) as submitted by the Nigerian government to the CRC.
Nigeria Child Survival Policy (2002)	Excluded	This document was a policy project on the rights of a child to life.
National Health Policy (2016)	Excluded	This policy document addresses health policies in Nigeria and nothing about disability.
National Guidelines on Promoting Access of Young People to Adolescent- and Youth-Friendly Services in Primary Health Care Facilities in Nigeria	Excluded	Guidelines for access to youth-friendly health services. Does not address disability.
Health policies and maternal/child health in Bayelsa state, Nigeria	Excluded	Study on the national trends in maternal and child health analyzed with a focus on health indicators which are not related to childhood disability.
Inclusive education for persons with disabilities in Nigeria: How far?	Excluded	Not a policy document; study investigating inclusive education practices as they affect persons with disability in Nigeria.
Convention of the Rights of the Child Report (2005)	Excluded	This report was incomplete as it included 5 report documents (concluding observation, list of delegates, state report, CRC statement, and summary report) as submitted to the CRC and was not the most recent submission from the Nigerian government to the CRC.
Convention of the Rights of the Child Report (1993)	Excluded	This report was incomplete as it included 3 report documents (concluding report, state report, and summary report) as submitted to the CRC and was not the most recent submission from the Nigerian government to the CRC.

**Table 5 ijerph-20-06996-t005:** CRC and CRPD article word frequency analysis.

CRC/CRPD Articles	Detail	Frequency
CRC Article 20	Children Deprived from Family	368
CRPD Article 23	Respect for Home and Family	136
CRPD Article 7	Children	97
CRC Article 23	Children with Disabilities	92
CRC Article 28	Education	82
CRC Article 31	Leisure/Play culture	80
CRPD Article 8	Awareness Raising	76
CRPD Article 6	Women and Girls	66
CRC Article 3	Best Interest of Child	36
CRPD Article 30	Cultural Life and Recreation	29
CRC Article 24	Health and Services	22
CRC Article 42	Knowledge of Rights	18
CRPD Article 24	Education	10
CRC Article 12	Respect to Views	3

**Table 6 ijerph-20-06996-t006:** CRC and CRPD articles included in coding framework.

CRC/CRPD Articles	Details	Coded Themes
CRC—Article 20	Children Deprived from Family	Assistance, Law
CRC—Article 23	Children with Disabilities	Law, Rights
CRC—Article 28	Education	Law
CRC—Article 31	Leisure, Play, and Culture	Law
CRPD—Article 23	Respect for Home and Family	Services
CRPD—Article 8	Awareness Raising	Accessibility, Awareness Raising, Law
CRPD—Article 7	Children	Accessibility, Awareness Raising

Coded theme factors associated with adherence to the CRC recommendations taken from the Nigerian policy documents and reports to the CRC.

**Table 7 ijerph-20-06996-t007:** Resulting theme and associated policy document/reports.

Theme	Details	Associating Articles/Policy Document/Reports
**I**	Participation	Nigerian Policy Documents and CRC Reports
**II**	Support Systems	Nigerian Policy Documents and CRC Reports
**III**	Awareness Raising	CRPD Articles 7 and 8
**IV**	Factors associated with adherence to the CRC	CRC Articles 20, 23, 28, and 31
**V**	Laws and Rights	CRC Articles 20, 23, 28, and 31; CRPD Article 8
**VI**	Services	CRPD Article 23; CRC Articles 20, 23, 28, and 31

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
