# Peer review of "A Portrait of the Rights of Children with Disabilities in Nigeria: A Policy Review"

_ijerph, 2023, doi:10.3390/ijerph20216996_

Round 1

Reviewer 1 Report

Thank you for this opportunity to review the article “A portrait of the rights of children with disabilities in Nigeria: A policy review.” 

The authors provided good introduction about the historical context of disability-related policies internationally. The authors also presented an overview of Nigeria’s policies and legislation pertaining to children with disabilities. They identified the research gap that the article seeks to address. The identified gap and the research questions can help readers to understand the purpose of this article. 

Below are the comments and suggestions I made: 

1.The authors should outline the document selection criteria clearly. Although the authors stated that unpublished reports and policies were excluded, it is unclear what other criteria were applied during the screening and selection process. I recommend presenting inclusion and  exclusion criteria in a table for clarity. 

2.I would suggest the authors write the analysis process clearly. Some questions I have in mind when I read “analysis and syntheses.”

1)    “One author proceeded with coding with regular discussions with the team on any new or diverging themes identified” which author? The first author? 

2)    “Other Nigerian Disability policy documents as well as the UN reports were coded by the first author with bi-weekly meetings with a second reviewer to review and validate the coding process, ensure agreement, and address discrepancies on the new emerging themes.” Does it mean only the first author coded Nigerian Disability policy documents as well as the UN reports? Who is the second reviewer? 

3)    “Every third Nigerian policy document was coded by two research team members to validate the emerging themes among investigators and differences were discussed and dealt with by mutual agreement” This is very confusing. What is “every third Nigerian policy document”? who are two “research team members”? Among “investigators,” who are investigators? 

4)    “Analysis of the codes was done by two investigators and critically debriefed with the other members of the team.”  Again, who are two investigators? And who other the other members of the team? 

I noticed varied terms like 'reviewer', 'author', 'investigators', and 'research team members' being used to describe similar roles. This inconsistency can lead to significant confusion for readers." Please keep the term consistently.            

3. I suggest the authors add a section to introduce researchers’ positionality, which refers to the position a researcher has chosen to adopt in a qualitative study. 

4. The authors mentioned “Table 6” and “Table 12” in the text, but where are the tables? I only see tables 1-5.

5. The discussion is too lengthy. As a reader, it is hard to grasp the points the authors tried to convey. I recommend that the authors center their discussion around the research questions. Additionally, I noticed repetitive mentions of areas lacking for children with disabilities and non-inclusive policies. Instead of reiterating these points, I suggest the authors use straightforward language to recommend future policy directions. The discussion section needs restructuring and major revision. 

Author Response

RE: A portrait of the rights of children with disabilities in Nigeria: A policy review

Thank you very much for taking the time to review this manuscript titled: A portrait of the rights of children with disabilities in Nigeria: A policy review. Please find the detailed responses below and the corresponding revisions/corrections in red and highlighted in yellow in the re-submitted files

Comment 1: The authors should outline the document selection criteria clearly. Although the authors stated that unpublished reports and policies were excluded, it is unclear what other criteria were applied during the screening and selection process. I recommend presenting inclusion and exclusion criteria in a table for clarity. 

Response 1: Thank you so much for your comment, we agree with your comment and have now addressed it in Lines 154 – 165, 168 – 180, and the Inclusion and exclusion criteria table (Table 3) has been included in Line 167

Comment 2: I would suggest the authors write the analysis process clearly. Some questions I have in mind when I read “analysis and syntheses.” 

  1. “One author proceeded with coding with regular discussions with the team on any new or diverging themes identified” Which author? The first author?
  2. “Other Nigerian Disability policy documents as well as the UN reports were coded by the first author with bi-weekly meetings with a second reviewer to review and validate the coding process, ensure agreement, and address discrepancies on the new emerging themes.” Does it mean only the first author coded Nigerian Disability policy documents as well as the UN reports? Who is the second reviewer?
  3. “Every third Nigerian policy document was coded by two research team members to validate the emerging themes among investigators and differences were discussed and dealt with by mutual agreement” This is very confusing. What is “every third Nigerian policy document”? who are two “research team members”? Among “investigators,” who are investigators?
  4. “Analysis of the codes was done by two investigators and critically debriefed with the other members of the team.”  Again, who are the two investigators? And who other the other members of the team?
  5. I noticed varied terms like 'reviewer', 'author', 'investigators', and 'research team members' being used to describe similar roles. This inconsistency can lead to significant confusion for readers." Please keep the term consistent.

Response 2: We appreciate this well-detailed review, and as such have addressed each of the comments in a point format below (Lines 194 – 209.)

  1. The first author (RE) proceeded with coding with regular discussions with the team on any new or diverging themes identified.
  2. Yes, the first author (RE) coded the Nigerian disability policy document as well as the UN reports. The second reviewer was NDO.
  3. “Every third Nigerian policy document was coded by two research team members to validate the emerging themes among investigators and differences were discussed and dealt with by mutual agreement” means that the “first author” who in this paper is RE coded each document and on every third document coded with NDO one of the coauthors. NDO in this paper is referred to as a member of the research team to ensure mutual agreement and consistency, and differences were taken to the other members of the research team (KS and LS) who are also coauthors (initially referred to as investigators) of the paper.
  4. The other two investigators are KS and LS who are coauthors on this paper. They are now referred to as members of the research team and not investigators to ensure consistency.
  5. The terms have now been made consistent. We took out the terms “reviewers” and “investigators” and used “author” when referring to one member of the team, and “research team” when referring to the other members of the team.

Comment 3: I suggest the authors add a section to introduce researchers’ positionality, which refers to the position a researcher has chosen to adopt in a qualitative study. 

Response 3: Thank you so much for pointing out this pertinent point, we have now addressed this comment and included it in Lines 210 - 225. 

Comment 4: The authors mentioned “Table 6” and “Table 12” in the text, but where are the tables? I only see tables 1-5.

Response 4: Thank you so much for pointing out this error on our part, we have now addressed this in the manuscript. Tables 1-7 can be seen in the manuscript in lines 129 (Table 1), 165 (Table 2), 166(Table 3), 237(Table 4), 248(Table 5), 249 (Table 6), 252 (Table 7).

Tables 8 – 13 (now tagged with Supplementary file in the manuscript) are included in the supplementary document that was submitted with the manuscript (Labelled IJERPH Policy review supplementary document) as the tables were too big to be included in the manuscript. An updated version of the supplementary document has also been submitted with this manuscript revision.

Comment 5: The discussion is too lengthy. As a reader, it is hard to grasp the points the authors tried to convey. I recommend that the authors center their discussion around the research questions. Additionally, I noticed repetitive mentions of areas lacking for children with disabilities and non-inclusive policies. Instead of reiterating these points, I suggest the authors use straightforward language to recommend future policy directions. The discussion section needs restructuring and major revision.

Response 5:  We agree with this comment and as such the discussion was reworked to incorporate reviewers’ suggestions and it has now been restructured to 2 pages as seen in Line 605 – 721.

Reviewer 2 Report

Thank you for the opportunity to review this important and interesting paper. Please see my comments and suggestions below.

It would be helpful to have more clarity on the inclusion and exclusion criteria for the documents, You state 14 documents met the criteria but Table 3 only has 6 included documents listed and 4 excluded. It would be useful to understand the reasons for exclusion of all 12 documents out of the 26 which were excluded.

It would be useful to have an explanation of how state and federal policies work in Nigeria to provide context for those not familiar with Nigeria.

It is unclear to me how the coded themes in table 5 relate to the parent and child nodes and Figure 1.

Line 491 the word "in" repeated twice.

Line 611 increase should not have a capital I.

The sentence starting with "This initiative..." on line 737 needs rewriting as it is unclear.

Line 742 nigerian needs a capital N.

Cultural resistance, attitudes to disability and stigma could perhaps be explored more in the discussion?

LOI should be included in the list of abbreviations.

I hope this work enables the lives of children with disabilities in Nigeria to be improved.

There are a few minor errors which need correcting.

Author Response

RE: A portrait of the rights of children with disabilities in Nigeria: A policy review

Thank you very much for taking the time to review this manuscript titled: A Portrait of the Rights of Children with disabilities in Nigeria: A Policy Review. Please find the detailed responses below and the corresponding revisions/corrections in red and highlighted in yellow in the re-submitted files.

Comment 1: It would be helpful to have more clarity on the inclusion and exclusion criteria for the documents. You state that 14 documents met the criteria, but Table 3 only has 6 included documents listed and 4 excluded. It would be useful to understand the reasons for the exclusion of all 12 documents out of the 26 that were excluded.

Response 1: We agree with this clarification request and the Inclusion and Exclusion criteria table (Table 3) included in Line 166.

Table 4 has been updated in line 237 with missing included policy documents (Nigerian Child Health Policy, 2006). There are 14 included documents (6 Nigerian policy documents are outlined in the table and the CRC reports included 8 documents) and 12 excluded documents which are also well outlined in Table 4. 

Comment 2: It would be useful to have an explanation of how state and federal policies work in Nigeria to provide context for those not familiar with Nigeria.

Response 2: We agree with this comment and as such have addressed it in Lines 62 - 67.

Comment 3: It is unclear to me how the coded themes in Table 5 relate to the parent and child nodes and Figure 1.

Response 3: We agree with this clarification request and as such made corrections as follows. Table 5 (now Table 6) was wrongly referenced and the correct table (Table 7) which relates to the parent and child node in Figure 1 (which can be found in the supplementary file) has now been included in Line 248.

Comment 4: Line 491 the word "in" repeated twice. 

Response 4:  We agree with this comment and have now addressed it in line 518

Comment 5: Line 611 increase should not have a capital I. 

Response 5: Although we agree with this comment, the paragraph was removed during the review of the discussion section. Hence the change wasn't necessary anymore.

Comment 6: The sentence starting with "This initiative..." on line 737 needs rewriting as it is unclear. 

Response 6: Although we agree with this comment, the paragraph was removed during the review of the discussion section. Hence the change wasn't necessary anymore.

Comment 7: Line 742 nigerian needs a capital N.

Response 7: Although we agree with this comment, the paragraph was removed during the review of the discussion section. Hence the change wasn't necessary anymore. 

Comment 8: Cultural resistance, attitudes to disability, and stigma could perhaps be explored more in the discussion?

Response 8: We agree with this comment and as such addressed it in Lines 701 -709.

Reviewer 3 Report

I am very pleased to be offered the opportunity to read this important paper. This paper offers important insight into the ways and extent to which the rights of children with disabilities are being implemented in law and policy in Nigeria. Doing so contributes to the existing field of study allowing for comparisons to be made on the implementation of disability rights for children across the world as well as in specific countries, and where gaps can arise. The paper reaches a key conclusion in that the intersectionality of disability and childhood is rarely explicitly considered or recognised in law and policy, leading to a potential undermining of the rights of children with disabilities. As such this makes an important contribution to the field. I do have some minor comments below which I feel will help further strengthen the paper. 

Pg 1, line 40 - 'advocates' - I suggest changing to 'affirms' as human rights law does more than advocate, but sets out rights and standards  countries need to abide by. 

Pg 1, line 45 - uncf should be unicef

Pg 2, unohchr - abbreviate to ohchr which is more commonly used and understood

Pg 3, line 102-104 - the intention is to actively monitor implementation of the human rights treaty rather than to 'provide standards'

Pg 4, line 143 - write out LOI abbreviation in full the first time it is mentioned

What were the limitations of the search strategy?

Table 3 has 10 documents listed but the search discussed above identifies 26 and 14 met the criteria - it is not clear why 10 documents are listed. Please clarify. 

Table 4 - Should CRC article 7 actually be CRPD article 7?

It would be helpful to have more clarification on how the themes were reached.  Were the CRC and CRPD articles explicitly mentioned in the documents or do they (perhaps inadvertently?) align? 

The discussion and conclusion summarises the findings. I feel this could be lifted further by discussing the contribution the paper makes to the field of study and making stronger links to the existing literature on the implementation of the rights of children with disabilities

Author Response

RE: A portrait of the rights of children with disabilities in Nigeria: A policy review

Thank you very much for taking the time to review this manuscript titled: A Portrait of the Rights of Children with disabilities in Nigeria: A Policy Review. Please find the detailed responses below and the corresponding revisions/corrections in red and highlighted in yellow in the re-submitted files.

Comment 1: Pg 1, Line 40 - 'advocates' - I suggest changing to 'affirms' as human rights law does more than advocate, but sets out rights and standards  countries need to abide by 

Response 1: We agree with this comment and as such addressed it in line 40.

Comment 2: Pg 1, Line 45 - uncf should be unicef 

Response 2: We agree with this comment and as such address it in line 45.

Comment 3: Pg 2, unohchr - abbreviate to ohchr which is more commonly used and understood.

Response 3: We agree with this comment and as such address it in Lines 21, 84, 87, 89, 90, 95, 159, 177, 787.

Comment 4: Pg 3, Lines 102-104 - the intention is to actively monitor the implementation of the human rights treaty rather than to 'provide standards'.

Response 4: We agree with this comment and as such address it in lines 107 - 109.

Comment 5: Pg 4, line 143 - write out the LOI abbreviation in full the first time it is mentioned.

Response 5: Thank you for pointing this out. However, LOI was initially written in full in Line 91, where it was mentioned for the first time. 

Comment 6: What were the limitations of the search strategy?

Response 6:

Thank you for pointing this out. We addressed this in Lines 162-164 and Lines 724-735.

And

The limitations of the search strategy are “Many Nigerian policy documents were also inaccessible and unavailable on-line.” And “In this policy review, most of the disability policies which were readily accessible following database and website search were federal related bills and just one state related policy (i.e., Lagos State Disability Bill, 2010). Despite later findings of state-related policies in CRC reports (table 13), the content of these documents remains inaccessible for review. Although the CRC and CRPD articles were used to form the mapping framework, the review was unable to map the Nigerian CRPD reports due to unavailability. This lack of report and access to policy documents may hinder policy implementation and the alignment with UN goals, particularly for children with disabilities, underscoring the pivotal advocacy role of health and education professionals serving these families [23] At the time of this policy review, there had been no updated cycle report from Nigeria since 2015 and no recent updates to the different policy documents.”

Comment 7: Table 3 has 10 documents listed but the search discussed above identifies 26 and 14 met the criteria - it is not clear why 10 documents are listed. Please clarify. 

Response 7: We agree with this comment and as such addressed it with the Inclusion and Exclusion criteria table (Table 3) included in Line 166.

Table 4 has also been updated in line 237 with missing included policy documents (Nigerian Child Health Policy, 2006). There are 14 included documents (6 Nigerian policy documents are outlined in the table and the CRC reports included 8 documents) and 12 excluded documents which are also well outlined in Table 4.

Comment 8: Table 4 - Should CRC article 7 actually be CRPD article 7?

Response 8: Thank you for pointing this error out. It has as such been addressed in Table 4 (Now Table 5  - Line 248). Yes, In Table 5 CRC article 7 is indeed in CRPD article 7 as properly outlined as included in Table 6.

Comment 9: It would be helpful to have more clarification on how the themes were achieved.  Were the CRC and CRPD articles explicitly mentioned in the documents, or do they (perhaps inadvertently?) align? 

Response 9: We agree with this comment and have now addressed it in lines 238 – 248. In response to the question, the CRC and CRPD documents were not always explicitly mentioned in the policy documents. However, the themes were achieved by using the CRC and CRPD articles as a framework of what is expected of the Nigerian disability and childhood disability policies creating a mapping framework (Line 239). Therefore, analyzing the policy document using these mapping frameworks generated more themes (Line 243) that aligned with the CRC and CRPD articles included in this policy review. 

Round 2

Reviewer 1 Report

The authors have responded to all the comments. I believe this paper is now ready for publication.